# Graphene-Based Far-Infrared Therapy Promotes Adipose Tissue Thermogenesis and UCP1 Activation to Combat Obesity in Mice

**DOI:** 10.3390/ijms26052225

**Published:** 2025-02-28

**Authors:** Jinshui Zhang, Shuo Li, Xin Cheng, Xiaocui Tan, Yingxian Shi, Guixin Su, Yulong Huang, Yang Zhang, Rui Xue, Jingcao Li, Qiongyin Fan, Huajin Dong, Yun Deng, Youzhi Zhang

**Affiliations:** 1Beijing Institute of Pharmacology and Toxicology, Beijing 100850, China; 2021201386@aust.edu.cn (J.Z.); shuoli1204@163.com (S.L.); chengxinhh2025@163.com (X.C.); tan135159@163.com (X.T.); syx15138911232@163.com (Y.S.); sgx07012024@163.com (G.S.); 18176702599@163.com (Y.H.); zhangyang@bmi.ac.cn (Y.Z.); hly19830718@sina.com (R.X.); lijing-cao00@126.com (J.L.); bingofqy@163.com (Q.F.); 13522700930@139.com (H.D.); 2School of Medicine, Anhui University of Science and Technology, Huainan 232001, China

**Keywords:** graphene-FIR therapy, HFD, obesity, AMPK, adipose tissue thermogenesis

## Abstract

Hyperthermia (HT) has broad potential for disease treatment and health maintenance. Previous studies have shown that far-infrared rays (FIRs) at 8–10 μm can potentially reduce inflammation, oxidative stress, and gut microbiota imbalance. However, the effects of FIR HT on energy metabolism require further investigation. To investigate the effects of graphene-FIR HT therapy on diet-induced obesity and their regulatory mechanisms in energy metabolism disorders. After 8 weeks of hyperthermia, mice fed standard chow or a high-fat diet (HFD) underwent body composition analysis. Energy expenditure was measured using metabolic cages. The protein changes in adipose tissue were detected by molecular technology. Graphene-FIR therapy effectively mitigated body fat accumulation, improved dyslipidemia, and impaired liver function while enhancing insulin sensitivity. Furthermore, graphene-FIR therapy increased V_O2_, V_CO2_, and EE levels in HFD mice to exhibit enhanced metabolic activity. The therapy activated the AMPK/PGC-1α/SIRT1 pathway in adipose tissue, increasing the expression of uncoupling protein 1 (UCP1) and glucose transporter protein four (GLUT4), activating the thermogenic program in adipose tissue, and improving energy metabolism disorder in HFD mice. In short, graphene-FIR therapy represents a comprehensive approach to improving the metabolic health of HFD mice.

## 1. Introduction

The escalating global health crisis related to obesity is a pressing concern, as evidenced by the World Obesity Federation’s 2023 World Obesity Map, which projects that by 2035, approximately 4 billion people worldwide will be classified as overweight or obese [1]. Obesity is a chronic metabolic disorder characterized by an imbalance between energy intake and expenditure, leading to excessive fat storage in the body [2].

Adipose tissue plays a pivotal role in regulating energy and nutrient balance, making it a prime target for anti-obesity strategies. This tissue includes white adipose tissue (WAT), brown adipose tissue (BAT), and beige adipose tissue. Different WAT subtypes have significant differences in metabolism and function, including inguinal white adipose tissue (iWAT), visceral white adipose tissue (vWAT), and epididymal white adipose tissue (eWAT). IWAT is responsible for storing and releasing lipids, while beige and brown adipocytes belong to thermogenic adipocytes that can consume nutritional energy and activate thermogenesis [3]. BAT is known for increasing energy expenditure through the uncoupling protein 1 (UCP1), which reportedly reduces body weight and improves metabolic disorders in obese individuals [4]. Current clinical approaches to weight loss include pharmacotherapy, surgical interventions, exercise, and physical therapy. However, the efficacy of lifestyle and behavioral modifications is often limited, necessitating the exploration of additional pharmacological, surgical, and innovative physical therapies. Approximately 27 anti-obesity drugs have been investigated for clinical applications, mainly by targeting various pathways in the central nervous system or peripheral organs to regulate energy intake (EI) or energy expenditure (EE) [5,6,7]. However, the side effects of weight-loss drugs, such as nausea, diarrhea, vomiting, constipation, and abdominal pain, limit their clinical application. Bariatric surgery has numerous limitations, including the inability to meet global healthcare demands, restricted patient eligibility, challenging postoperative recovery, and high economic costs [8]. Most drugs approved by the Food and Drug Administration for obesity treatment have undesirable side effects [9]. Given these challenges, the focus has shifted towards discovering methods to boost energy expenditure and curtail energy intake.

The light wave around 8–14 µm is the far-infrared ray (FIR), which has strong radiation and permeability and is conducive to energy transfer to tissues, exciting molecular vibrational energy levels and improving body metabolism [10]. Therefore, FIR therapy is becoming a promising technology because it is noninvasive, and its effect is better than that of traditional medicine and surgical treatment. It has been proven to improve metabolic diseases, such as obesity, osteoporosis, and heart failure, in human and animal models [11,12,13]. Graphene, a new nanomaterial, has high electrothermal conversion efficiency and excellent optical properties. A heating film made of graphene has high FIR emissivity. When a bias voltage is applied, a large part of the electron energy can be converted into infrared radiation, and the peak of the radiation wavelength is about 8 μm, which can almost overlap with the infrared spectrum of the human body to form the same frequency resonance. It accelerates the body’s metabolism and blood flow and has a better therapeutic effect [14]. Graphene devices can improve skin damage and enhance the effects of hyperthermia [15]. Our previous study found that graphene-FIR therapy can induce weight loss caused by a high-fat diet (HFD) in mice without reducing energy intake, improve HFD-induced anxiety behavior, and regulate intestinal permeability and inflammatory response. The effect is better than ordinary carbon fiber-FIR therapy [16]. Metabolic modulators, such as the PPAR agonist GW501516 and the AMP-activated protein kinase (AMPK) agonist, known as “exercise mimetic agents”, have been shown to mimic the lipid-lowering effects of aerobic exercise, tricking the body into requiring a higher metabolism to help the body lose weight or improve endurance. Exercise mimetics, compounds designed to replicate exercise benefits, have faced safety issues and ethical controversies. While initially promising for metabolic enhancement, their misuse in sports has resulted in health complications and subsequent bans by athletic authorities, significantly hindering therapeutic development. In our previous study, graphene-FIR therapy effectively modulated gut microbiota homeostasis and activated AMPK, enhancing physical performance without the side effects associated with drug interventions [14]. It has potential utility as a mimetic agent in treating metabolic disorders. However, the mechanism by which graphene-FIR therapy affects fat reduction, particularly the regulation of energy metabolism, remains unclear.

AMPK and SIRT1 are the main sensors of energy metabolism and key factors regulating metabolic homeostasis. They maintain energy balance by activating catabolic pathways, enhancing mitochondrial oxidative metabolism and mitochondrial biogenesis, and increasing the expression and activity of PGC-1α [17]. Aldometanib, an AMPK agonist, has been reported to reduce blood glucose levels, delay NAFLD, and prolong the lifespan of *Caenorhabditis elegans* while also improving metabolic and cardiac dysfunction and exercise endurance [18]. Several studies have shown that activating AMPK/SIRT1 can improve metabolic disorders and ensure energy homeostasis in HFD mice [19,20]. Furthermore, studies on activating the AMPK/SIRT1 and BAT/beige pathways and the species changes in the gut microbiota have provided key mechanistic insights for anti-obesity [21,22]. In the present study, we investigated the effects of graphene-FIR therapy on HFD-induced obesity and metabolic disorders. Further, graphene-FIR therapy activated the thermogenic program in adipose tissue, increasing the expression of UCP1 and glucose transporter protein four (GLUT4), activating the AMPK/PGC-1α/SIRT1 pathway in iWAT and BAT, and improving energy metabolism disorder in HFD mice.

## 2. Results

### 2.1. Materialistics of Graphene and Carbon Fibers

A heating film with a single-layer graphene structure had better optical and electrical properties than a carbon fiber heating film. First, we confirmed the material characterization of graphene films and carbon fibers. Monolayer graphene is grown on copper foil and polyethylene terephthalate (PET) film by CVD (Appendix A), while carbon fiber-based devices are commercially available (Appendix A) and powered by an independent power supply (Appendix A). The SEM results showed that the graphene film was a two-dimensional film (Figure 1A), and the carbon fibers existed in the form of fibers with a graphitized structure (Figure 1C). Raman spectroscopy verified the single-layer nature of the graphene film (Figure 1B) and the multi-layer nature of the carbon fiber (Figure 1D). At the same time, we compared the emission peak of the two materials with the absorption spectrum of the human body and found that the highest emission peak position of the graphene device basically coincided with the characteristic absorption peak of the human body, while the emission peak wavelength of the carbon fiber device was lower than 8 µm (Figure 1E–G). These results suggest that FIR derived from graphene may act more efficiently to produce unique biological effects in the human body.

### 2.2. Graphene-FIR Therapy Can Increase Body Temperature and Blood Flow

The IR thermography results indicated that the surface temperature of the adult scapula and back increased after graphene-FIR therapy and remained elevated for a longer duration (Figure 2A). The animal experiments also showed that the surface and rectal temperatures of the Gra-FIR group were significantly higher than those of the Car-FIR group (Figure 2B–F). BAT and iWAT blood flow in Gra-FIR mice increased significantly compared with that in the Car-FIR group (Figure 2G–I). These results indicate that graphene-FIR therapy promotes adipose tissue thermogenesis and may have specific biological effects on obesity.

A schematic of the experimental procedure is shown in Figure 3A,B. An HFD can cause changes in the weight, shape, and related indicators of organs in mice. After 16 weeks of an HFD, compared to the control group, the body weights of the mice in the HFD group increased significantly (*p* < 0.05, *p* < 0.01, Figure 3C), indicating that the obese mouse model was successfully established. Compared with the model group, the body weight of mice in the graphene-FIR group decreased significantly, while there was no significant decrease in the Car-FIR group (Figure 3C; NC group: 29.8 ± 0.512 g; HFD group: 37.31 ± 1.327 g; HFD+Gra group: 31.04 ± 0.539 g; and HFD+Car group: 33.80 ± 1.156 g). Body composition analysis showed that compared with the model group, mice in the graphene-FIR group had reduced fat content and significantly increased muscle content (*p* < 0.05, *p* < 0.01, Figure 3D). Compared to the HFD group, the graphene-FIR group showed significantly reduced liver, iWAT, and kidney weights (*p* < 0.05, *p* < 0.01, Figure 3E,F). HE staining of the liver and adipose tissue showed that graphene-FIR therapy significantly reduced liver lipid droplet deposition and adipocyte hypertrophy in HFD mice (Figure 3G). Cold-exposed mouse models can be used to assess the activation status of adipose tissue. The results revealed that graphene-FIR therapy significantly increased BAT mass and reduced iWAT mass (*p* < 0.05; Appendix A), decreased the diameter of adipose tissue cells, increased the number of adipose tissue cells (Appendix A), increased BAT and iWAT blood perfusion (*p* < 0.01, Appendix A), and increased the body surface temperature (*p* < 0.01, Appendix A). Notably, the effects were better than those of the Car-FIR therapy. These results indicate that graphene-FIR therapy can prevent HFD-induced obesity and reduce body fat content.

### 2.3. Graphene-FIR Therapy Can Improve Dyslipidemia and Insulin Resistance in HFD Mice

An HFD usually leads to concurrent weight gain and metabolic disorders in mice [23,24]. The results showed that compared with the HFD group, graphene-FIR therapy significantly reduced TG, TC, HDL-C, and LDL-C in the mice serum (*p* < 0.05, *p* < 0.01, Figure 4A–D). Compared to the HFD group, graphene-FIR therapy significantly reversed the increase in liver function indices in HFD mice (*p* < 0.05, *p* < 0.01, Figure 4E,F), whereas Car-FIR therapy showed no significant difference. These results indicate that graphene-FIR therapy can effectively improve the abnormal serum lipid content and impaired liver function in HFD mice.

We found that graphene FIR therapy did not enhance glucose uptake or insulin resistance in normal mice (Appendix A). In the GTT, graphene-FIR therapy significantly reversed the increase in the fasting blood glucose value of HFD mice, and the AUC value was lower than that of the model group; however, there was no significant difference in the Car-FIR group (*p* < 0.01, Figure 4G,H). In the ITT, the blood glucose value and AUC of the graphene-FIR group were significantly lower than those of the HFD group, and the Car-FIR group did not show significant improvements (*p* < 0.05, *p* < 0.01, Figure 4I–K). We also found that serum insulin and leptin levels of HFD mice decreased, and the adiponectin level increased significantly after graphene-FIR radiation therapy compared to the HFD group (*p* < 0.01, Figure 4L–N). The homeostatic model assessment of IR (HOMA-IR) was used to evaluate the degree of insulin resistance. Graphene-FIR therapy significantly reversed the increase in HOMA-IR in the model group (*p* < 0.01, Figure 4O), proving that graphene-FIR therapy could improve insulin resistance caused by HFD. In the cold exposure model, we found that graphene-FIR therapy significantly reduced the AUC value of the IPGTT (*p* < 0.05; Appendix A), increased glucose uptake by the adipose tissue (*p* < 0.01, Appendix A), and improved glucose tolerance in mice. These results suggest that graphene-FIR therapy can alleviate impaired glucose tolerance caused by HFD, enhance insulin sensitivity, and improve glucose metabolism disorders.

### 2.4. Graphene-FIR Therapy Has No Impact on Energy Intake but Increases EE in HFD Mice

Usually, HFD causes an energy imbalance [18]. We evaluated the effects of graphene-FIR therapy on the EI and EE. The results showed that graphene-FIR therapy did not increase food or energy intake (Figure 5A–C). We hypothesized that the effect of graphene-FIR therapy on weight loss in mice may be caused by an increase in EE. The metabolic detection results revealed that after graphene-FIR therapy, oxygen consumption (V_O2_; Figure 5D,E), exhaled carbon dioxide (V_CO2_; Figure 5F,G), and EE (Figure 5H,I) were higher than those of the model group at all times (*p* < 0.05, *p* < 0.01). The respiratory exchange rate (RER), an indicator of metabolic fuel preference, did not differ between the groups (Figure 5J,K). We also examined the rectal temperature of the mice after 4 h at 4 °C and found that graphene-FIR therapy could maintain a higher rectal temperature (*p* < 0.05, *p* < 0.01, Figure 5L), indicating that graphene-FIR therapy could maintain enhanced non-shivering thermogenesis by burning fat. Moreover, we adjusted the metabolic rates (MRs) in the four groups that differed in body mass and composition [25] and observed that graphene-FIR therapy significantly increased the predicted MR, indicating the mechanical effect of graphene-FIR therapy on thermogenesis (*p* < 0.05, *p* < 0.01; Figure 4M–O).

Spontaneous activity for 24 h was used to examine the activity of the HFD mice. The results showed that graphene-FIR therapy could significantly reverse the decline in exercise volume of HFD mice during the day (7 AM to 7 PM), night (7 PM to 7 AM), and the whole day (*p* < 0.05, *p* < 0.01, Figure 5Q–T); however, the improvement effect of the Car-FIR group was not different from that of the model group. Therefore, graphene-FIR therapy restored activity in HFD mice.

### 2.5. Graphene-FIR Therapy Can Activate Adipose Tissue Thermogenic Program and Increase AMPK/SIRT1/UCP1 Protein Expression

As a sensor of cellular energy metabolism, AMPK plays an important role in the body’s energy regulation. It is also the master switch in body metabolism and is an important target for treating obesity caused by HFD [17]. Compared with the HFD group, graphene-FIR therapy could reverse the HFD-induced reduction in the protein expression of *P*-AMPK/AMPK, PGC-1α, SIRT1, UCP1, and GLUT4 (*p* < 0.05, *p* < 0.01, Figure 6A–E). Car-FIR therapy had no significant effect. We used RT-PCR to detect the expression of thermogenic genes in the iWAT of HFD mice. Graphene-FIR therapy increased the expression of *PGC-1α*, *PPARγ*, *Cidea*, *Dio2*, and *Prdm16* in the iWAT of HFD mice (*p* < 0.01, Figure 6F–J). The cold exposure model is another model to evaluate the thermogenic activation of adipose tissue. In the cold exposure model, graphene-FIR therapy could significantly increase the protein expression of P-AMPK/AMPK, PGC-1α, UCP1, and GLUT4 (*p* < 0.05, *p* < 0.01, Appendix A) and the mRNA levels of thermogenic genes (*p* < 0.05; Appendix A) and activate the thermogenic program of BAT. These effects could be blocked by an AMPK inhibitor (*p* < 0.05, Appendix A). Therefore, graphene-FIR therapy could activate adipose tissue thermogenesis, activate the AMPK/SIRT1/PGC-1α pathway, increase UCP1 and GLUT4 protein expression, improve glucose uptake in adipose tissue, induce WAT “browning”, and enhance energy consumption.

## 3. Discussion

Owing to the increasing prevalence of obesity and the fact that most therapeutic drugs indirectly act on the central nervous system to suppress appetite or gastrointestinal fat absorption to achieve weight loss goals, side effects are significant and prone to rebound [26,27]. In this study, we developed an FIR photothermal therapy based on a graphene heating film, which acts on the whole body to improve obesity status (fat browning, increased body heat production, and increased energy consumption). This method ensures efficient and low side-effect weight loss efficacy and regulates fat metabolism, which has a more significant effect than carbon fiber material thermal therapy.

Previous studies have reported that a long-term HFD can cause weight gain and abnormal blood lipid levels in mice [28]. In this study, the phenotype of obese mice induced by an HFD was consistent with most reports, indicating the successful establishment of the model. Body and fat weight are important indicators of obesity [29]. The experimental results showed that graphene-FIR therapy could effectively improve body weight, organ weight (iWAT, liver), serum lipid levels, and liver function indicators (ALT and AST) in mice. Although the serum HDL-C levels in the graphene-FIR therapy groups in the present study showed a downward trend, the values were stable at normal levels, and this change may have occurred because the lipid profile of mice is different from that of humans [30]. Long-term HFD-induced obesity can lead to severe insulin resistance in the body, thereby causing an increase in blood glucose and insulin levels [31]. In this study, graphene-FIR therapy significantly reduced blood glucose and insulin levels in HFD mice, reduced the HOMA-IR index, and reduced AUC values in GTT and ITT. In contrast, carbon fiber-FIR therapy did not improve these indicators. This suggests that graphene-FIR therapy can regulate the metabolic state of HFD mice by enhancing their insulin sensitivity.

Studies have shown that leptin contributes to energy balance and metabolic regulation [32]. Obese patients have high levels of leptin and are not sensitive to exogenous leptin. The iWAT tissue of obese mice can secrete excessive leptin, resulting in leptin resistance in the body. The circulating level of leptin is directly proportional to the content of adipose tissue, which is manifested as a decreased ability to suppress appetite, increased intake, and increased body weight [33]. Adiponectin, which is secreted by adipocytes, regulates glucose and lipid levels in serum and reduces inflammation levels in the body [34]. The serum level of adiponectin is inversely proportional to the total body fat content and is one of the highest concentrations of hormones in plasma. Weight loss or food restriction in obese patients increases circulating adiponectin levels, which are associated with increased insulin sensitivity. Serum adiponectin concentration is lower in patients with hyperinsulinemia and type 2 diabetes than in normal subjects [35]. Our findings showed that graphene-FIR therapy improved serum leptin levels and decreased adiponectin levels in HFD mice.

Reducing the body’s energy intake or increasing EE are reliable weight loss strategies [36]. Currently, most weight loss drugs are aimed at treating obesity by reducing EI [9]. This study showed that neither graphene-FIR nor carbon fiber-FIR therapy affected the food and EI of HFD mice, indicating that they did not reduce body weight by reducing EI. Energy metabolism testing revealed that graphene-FIR therapy significantly increased V_O2_, V_CO2_, and EE in HFD mice. This indicates that graphene-FIR therapy may improve obesity in HFD mice by promoting EE. These effects on weight are mainly achieved by activating BAT thermogenesis and promoting WAT browning, leading to enhanced EE. Additionally, we report the specific response of fat storage to graphene-FIR therapy in terms of heat production and fat breakdown. Graphene-FIR therapy increased the mRNA expression of thermogenesis and lipolysis in the BAT and iWAT of HFD mice and the cold-exposed mice model.

The thermogenic ability of BAT is impaired in obese patients, and their resistance to cold stress is worse than that of normal people. This study found that graphene-FIR group mice can significantly increase the body temperature of cold-exposed mice, indicating that graphene-FIR therapy may enhance the resistance to cold stress by promoting body heat production. We demonstrated this in both the HFD mice model and the cold exposure model. Meanwhile, in the cold exposure model, we found that graphene-FIR therapy could increase adipose tissue (BAT and iWAT) blood perfusion and the body surface temperature of mice. This effect is not observed in carbon fiber-FIR therapy.

Activating the UCP1-dependent thermogenic pathway in BAT and iWAT can promote EE in the body and improve HFD-induced obesity [37]. Graphene-FIR therapy significantly increased the expression levels of UCP1 protein in BAT and iWAT. Reports suggest that AMPK and SIRT1, as two important nutritional and energy sensors, independently or synergistically increase the expression and phosphorylation of PGC-1α. PGC-1α also interacts with various transcription factors to stimulate mitochondrial metabolism. Moreover, AMPK and SIRT1 also have a bidirectional regulatory role, increasing the expression of other thermogenic factors [38,39,40]. Therefore, PGC-1α, SIRT1, and AMPK form a metabolic regulatory network to control the body’s energy metabolism. AMPK-SIRT1 in adipose tissue can regulate glucose homeostasis and insulin sensitivity. This study found that graphene-FIR therapy could activate the thermogenic program in adipose tissue, increasing the protein expression of AMPK/SIRT1/PGC-1α in adipose tissue (iWAT and BAT) and improving metabolic disorders in HFD mice. In the cold exposure model, we also found that graphene-FIR therapy could activate BAT and increase UCP1 expression through the AMPK/PGC-1α pathway. At the same time, after the administration of an AMPK inhibitor in mice, the effect of graphene-FIR treatment on thermogenesis and blood flow acceleration in mouse adipose tissue was inhibited, indicating that graphene-FIR treatment mainly acted through the AMPK signaling pathway. GLUT4, a member of the glucose transporter family, is a membrane protein that exists in skeletal muscle, liver, adipose tissue, and myocardium. Its ability to translocalize is the key to the regulation of glucose metabolism and is closely related to the occurrence of metabolic diseases such as obesity, T2DM, hyperlipidemia, and atherosclerosis. At the same time, it has been reported that the ability of adipose tissue and liver to uptake glucose and the expression of GLUT4 protein in high-fat diet mice are decreased [41]. Consistent with the literature description, our experiment found that the expression of GLUT4 protein in the adipose tissue of mice in the HFD group was significantly decreased, and graphene-FIR therapy could improve the decrease in GLUT4 protein, enhance their glucose uptake ability, and improve their metabolic state.

Our study had some limitations. Our study did not exclude other mechanisms of anti-obesity effects of Gra-FIR therapy (regulatory effects on the liver, hypothalamus, and muscle tissue). Meanwhile, we proposed a new treatment strategy, which is using Gra-FIR therapy to improve obesity. Graphene-FIR therapy activated the thermogenic program in adipose tissue, increasing the expression of UCP1 and GLUT4, activating the AMPK/PGC-1α/SIRT1 pathway in BAT and iWAT, and improving energy metabolism disorder in HFD mice.

## 4. Materials and Methods

### 4.1. Materials

Details of the materials are provided in the Appendix A (including manufacturers and product numbers).

### 4.2. Mice Experiments

The Committee on Ethics of Animal Experiments of the Beijing Institute of Pharmacology and Toxicology (Beijing, China) approved this study (approval number: IACUC-DWZX-2022-632). All the institutional and national guidelines for the care and use of laboratory animals were followed. Appropriate measures were taken to minimize the number and suffering of animals.

C57BL/6J male mice (SPF Biotechnology Co., Ltd., Beijing, China) were used in the experiments, and they were 6–8 weeks old. The mice were subjected to a 12 h light/12 h dark cycle and were kept in a temperature- and humidity-controlled environment (24 ± 2 °C, 55 ± 5% relative humidity). After adapting to the feeding environment for 1 week, the animals were randomly divided into a control group (NC, *n* = 10) and an HFD group (HFD, *n* = 30) according to their body weight. They were fed a conventional diet or an HFD for 16 weeks (Appendix A).

After 16 weeks, the mice were weighed and selected to meet the criteria for an HFD (inclusion criterion: body weight of the mice was 20% higher than the average body weight of the control mice fed a normal diet). The mice that met the criteria were divided into an HFD group, an HFD + Gra group, and an HFD + Car group, with 7 mice in each group. Body weight and food and energy intake were measured weekly.

Animal protocol 1: Mice were placed in 40 × 25 × 20 cm rectangular plastic boxes for FIR irradiation intervention. The FIR intervention box contained 16 FIR slices (graphene or carbon fiber) in series with a single resistance value of 27 Ω. The FIR film was mounted on the inner surface of a box and powered using an independent variable voltage power supply (APS-2002D, AVITECH INSTRUMENT Co., Ltd., Shenzhen, China). For the FIR intervention experiments, mice were placed in the inner center of the box and allowed to move freely. The temperature in the control chamber was about 24.0 ± 1 °C, and the temperature in the FIR chamber was about 30.0 ± 1 °C. The temperature of the laboratory was controlled at about 25 °C. The input power of the intervention box was set to 30 W for both the graphene and carbon fiber. Irradiated mice (Gra and Car groups) were treated with FIR twice daily for 2 h for 8 weeks.

Cold-exposed mouse model establishment and treatment: The mice in the treatment group were given far-infrared intervention for 7 days (the same way as the HFD mice treatment group), and the mice were placed in a 4 °C intelligent artificial climate chamber. The mice were exposed to cold for 4 h a day for a total of three days to establish the mouse model of intermittent cold exposure, and then behavior detection and sampling were performed within a certain period of time.

The mice were fasted for 12 h on the day before sampling, after which the heart, liver, spleen, kidney, BAT, iWAT, colon, and feces were collected for subsequent testing.

### 4.3. Hyperthermia Therapy in Human Subjects

This study was conducted in accordance with the ethical guidelines of the Beijing Institute of Pharmacology and Toxicology (Beijing, China), and all subjects signed informed consent (AF/SC-08/02.391). The device of graphene optical wave therapy room is provided by Grahope New Materials Technologies Inc. (Shenzhen, China) (product model: FGB-1400, has completed the registration of medical devices in China, registration number: Guixie Zhuzhun 20212090236). Detailed experimental procedures are described in the Appendix A.

Subjects were acclimated in the test chamber 1 h before baseline imaging. After baseline imaging, participants’ supraclavicular fat stores (around the upper shoulder/upper back region) were exposed to a graphene-FIR phototherapy chamber for 45 min, during which thermal imaging photographs were taken. Subjects had thermal images of their supraclavicular fat depots taken 20 min after stopping FIR treatment. After that, the exact same experimental protocol was repeated for each volunteer.

### 4.4. Glucose Tolerance Test (GTT) and Insulin Resistance Test (ITT)

The GTT and ITT were performed as previously reported [42]. In short, GTT, mice fasting 12 h after intraperitoneal injection of 20% glucose solution (2.0 g/kg). For ITT, mice were fasted for 6 h and then intraperitoneally injected with insulin (0.75 U/kg). After that, the blood glucose meter was used to detect the blood glucose values at 0 min, 30 min, 60 min, 90 min, and 120 min, the blood glucose line chart was drawn, and the AUC under the curve was calculated.

### 4.5. Cold-Induced Thermogenesis

The mice were brought into a room with a laboratory temperature of about 25 °C for 2 h, and then the basal rectal temperature was measured. Then, the mice were placed in an intelligent artificial climate chamber for acute cold exposure at 4 °C for 4 h, and the rectal temperature of the mice was measured.

### 4.6. Body Composition Analysis

Body composition (fat, muscle, free water) was measured by Awake Small Animal Body composition analyzer. The mice were not anesthetized or killed during the whole experiment and were kept awake for detection. The mice were fasted for 12 h before monitoring; after weighing the body weight of the mice, the mice were placed in the machine detection tube in turn, and the parameters related to the fat content, muscle content and free water content of the mice were obtained after waiting for about 15 s.

### 4.7. Determination of EE by Indirect Calorimetry

The energy metabolism of mice was detected by the EM-XM energy metabolism detection system from Shanghai Tawang Company (EM-16M-WAG, Shanghai TOW Intelligent Technology Co., Ltd., Shanghai, China). During the intervention period, mice in each group were placed in metabolic cages to acclimate to the environment for 2 days, and then the formal detection began. During the formal detection, mice in each group were placed in order, single cage and single animal were detected, and the feeding environment was kept in a 12 h light/dark cycle. The system was turned on for gas balance detection for 1 h, and then the gas changes (oxygen consumption, carbon dioxide production, energy consumption) and spontaneous activity (movement distance and number) of mice were detected in real time. Each mouse was detected for 24 h, and then the data were collected for statistical analysis. Mass-adjusted metabolic rate (MR) ratios were tested using global regression-based analysis of covariance (ANCOVA) using body mass as a covariate.

### 4.8. Biochemical Analyses

The Triglyceride (TG), Total cholesterol (TC), Low-density lipoprotein cholesterol (LDL-C), High-density lipoprotein cholesterol (HDL-C), Glutamic oxalacetic transaminase (AST), Glutamic-pyruvic transaminase (ALT) levels in the mice serum were detected using an automatic biochemical analyzer (AU400, Olympus, Tokyo, Japan).

### 4.9. Tissue Staining

Tissue staining was performed as previously reported [11], and detailed steps for HE and Oil Red O staining can be found in the Appendix A.

### 4.10. Enzyme-Linked Immunosorbent Assay (ELISA)

Serum concentrations of adiponectin, leptin, and insulin were measured according to the ELISA kit manufacturer’s instructions.

### 4.11. RNA Extraction and Quantitative Real-Time PCR

SYBR Green I real-time PCR was used to detect mRNA transcription levels in white adipose tissue samples. Primer sequences are shown in Appendix A.

### 4.12. Western Blot

A total of 20 μg of total protein (BAT, iWAT) was wet-transferred to a PVDF membrane using SDS-PAGE (Milipore, Burlington, MA, USA). The membrane was incubated with 5% skim milk for 2 h at 25 °C. Then, *P*-AMPK (1:1000), AMPK (1:1000), β-actin (1:5000), PGC-1α (1:500), UCP1 (1:5000), SIRT1 (1:1000), GLUT4 (1:1000), and primary antibodies were used and incubated for 10 h at 4 °C in a shaker. They were then incubated with the corresponding secondary antibodies (1:5000) for 1 h at 25 °C. Subsequently, the developer solution uniformly covered the bands. Bands were visualized using the Protein Simple imaging system (FluorChem E 5.0.4, FE1159, Protein Simple Technology Co. Ltd., San Jose, CA, USA) and analyzed using ImageJ software (Version 1.53t, National Institutes of Health, Bethesda, MD, USA). Detailed information is provided in the Appendix A.

### 4.13. Statistical Analysis

All results are presented as the mean ± SEM. GraphPad Prism 7.5.1 and Image J software were used for statistical analysis and mapping. Student’s test was used to compare two groups; one-way ANOVA and Tukey’s post hoc test were used to compare more than two groups. Statistical significance was considered at *p* < 0.05.

## 5. Conclusions

Our results elucidate the various interrelated molecular mechanisms underlying graphene-FIR therapy. This study provides an effective technical strategy for hyperthermia to combat energy metabolism disorders and obesity and provides a preliminary explanation for the unique effect of far-infrared hyperthermia on fat reduction.

## Figures and Tables

**Figure 1 ijms-26-02225-f001:**
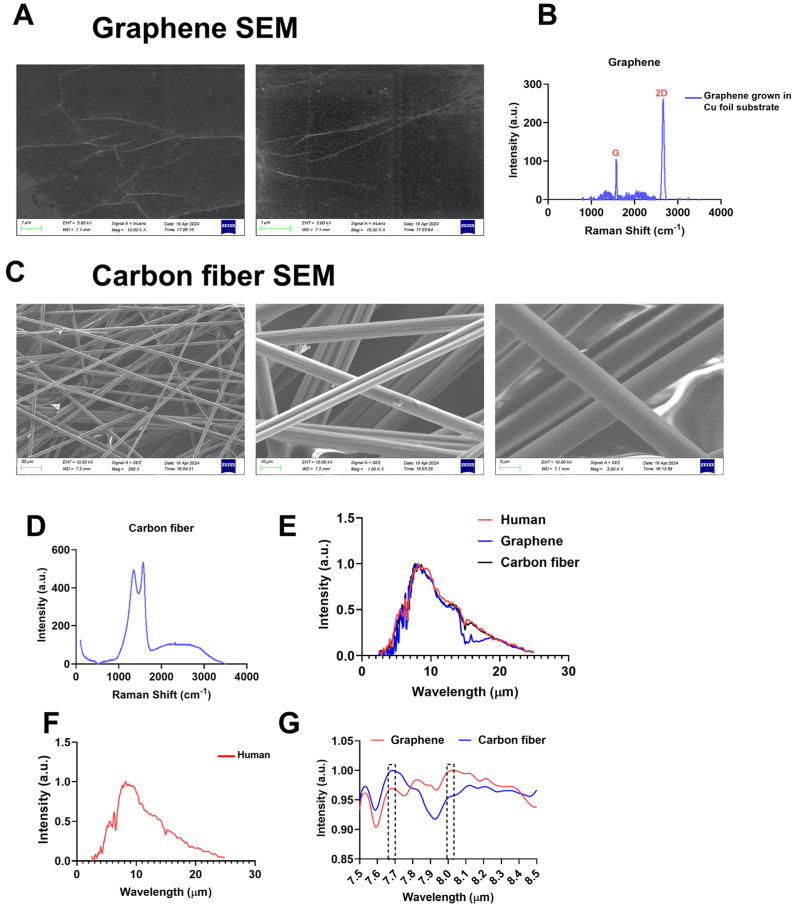
Specific FIR generated by graphene-based devices. (**A**) SEM image of graphene film; (**B**) Raman spectrum of the graphene FIR; (**C**) SEM image of carbon fiber film; (**D**) Raman spectrum of the carbon fiber FIR; (**E**) FIR emission spectra of graphene and carbon fiber (graphene: 30 W, carbon fiber: 30 W); (**F**) the absorption peak of the human body surface was approximately 8.0 µm; (**G**) the FIR emission peak of the graphene was approximately 8.03 µm, and the carbon fiber was approximately 7.685 µm.

**Figure 2 ijms-26-02225-f002:**
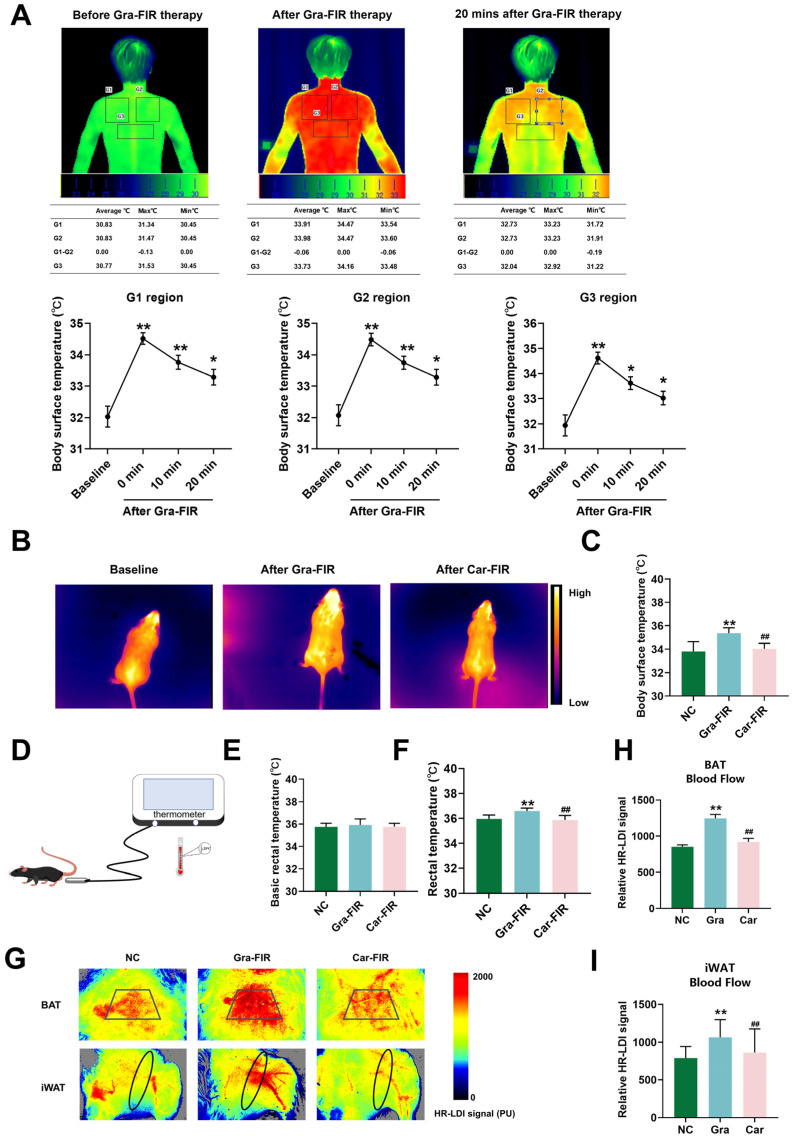
Graphene-FIR therapy induces thermogenesis and increases blood flow in vivo. (**A**) Schematic representation of thermal imaging of human supraclavicular area and temperature changes before or after Gra-FIR therapy. Data are displayed as mean ± SEM; *n* = 10, * *p* < 0.05, ** *p* < 0.01 VS baseline. The selected part of the black box is the temperature detection area. (**B**,**C**) Schematic diagram of thermal imaging and statistical analysis of temperature changes in mice after Gra-FIR or Car-FIR. (**D**–**F**) A schematic diagram of rectal temperature measurement in mice and statistical analysis of rectal temperature changes before and after the Gra-FIR or Car-FIR therapy. (**G**–**I**) Representative images and statistical analysis of blood perfusion in BAT and iWAT of mice after Gra-FIR or Car-FIR therapy. The site selected by the black circle box was the blood flow detection area. Data are displayed as mean ± SEM; *n* = 7, ** *p* < 0.01 VS NC group; ^##^
*p* < 0.01 VS Gra-FIR group.

**Figure 3 ijms-26-02225-f003:**
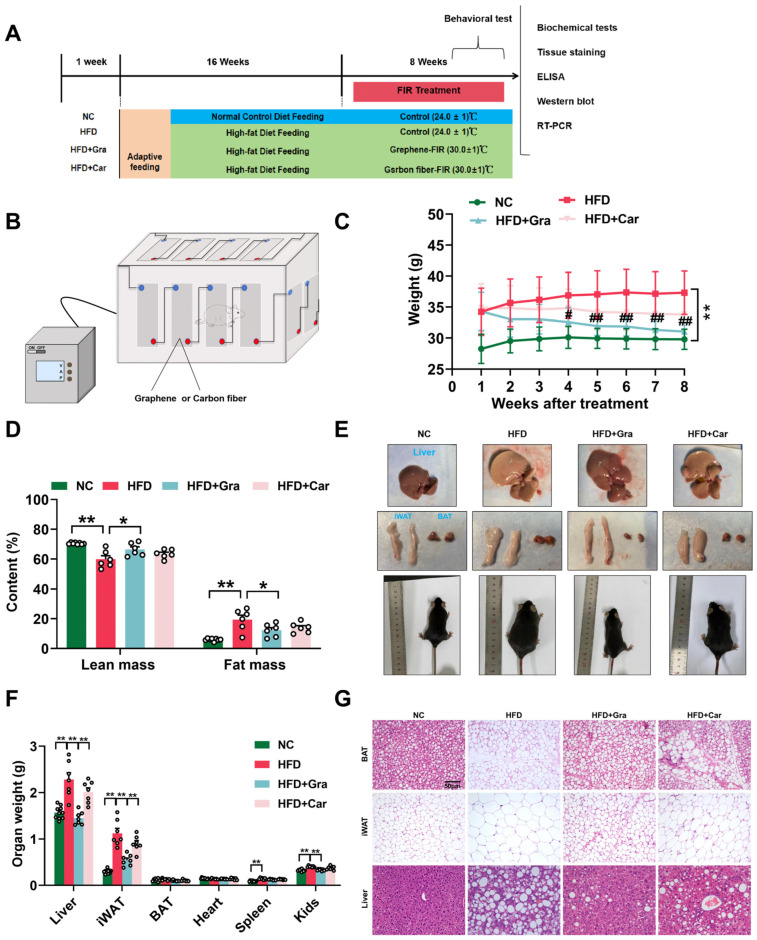
Graphene-FIR therapy can improve HFD-induced weight gain. (**A**) Schematic of the experimental design. (**B**) Schematic diagram of the FIR device. (**C**) Line chart of mice body weight in each group (*n* = 7–10). (**D**) The total body fat and mass content of each group (*n* = 6). (**E**) Representative images of liver, BAT, and iWAT in each group. (**F**) Organ weight in each group (*n* = 7–10). (**G**) HE staining of liver, BAT, and iWAT (200X; *n* = 3). Scale-bar: 50 μm. Data are displayed as mean ± SEM. * *p* < 0.05, ** *p* < 0.01 VS NC group; ^#^
*p* < 0.05, ^##^
*p* < 0.01 VS HFD group.

**Figure 4 ijms-26-02225-f004:**
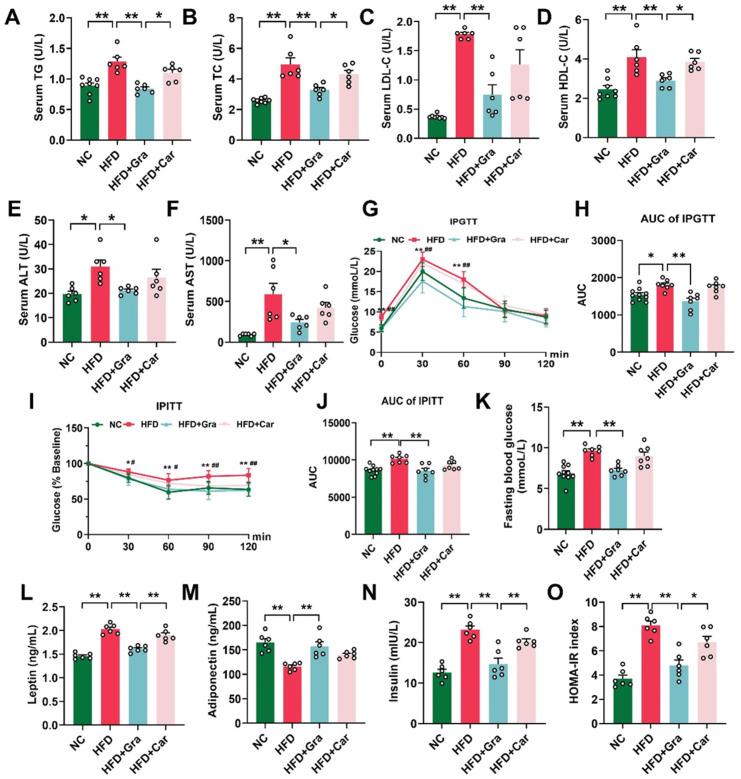
Graphene-FIR therapy can improve dyslipidemia and insulin resistance in HFD mice. (**A**–**F**) The serum levels of TG, TC, LDL-C, HDL-C, ALT, and AST. (**G**,**H**) GTT and area under the curve. (**I**,**J**) ITT and area under the curve. (**K**) Fasting blood glucose levels. (**L**–**N**) Serum levels of leptin, insulin, and adiponectin. (**O**) HOMA-IR index. Data are displayed as mean ± SEM; *n* = 6–10, * *p* < 0.05, ** *p* < 0.01; ^#^
*p* < 0.05, ^##^
*p* < 0.01 VS HFD group.

**Figure 5 ijms-26-02225-f005:**
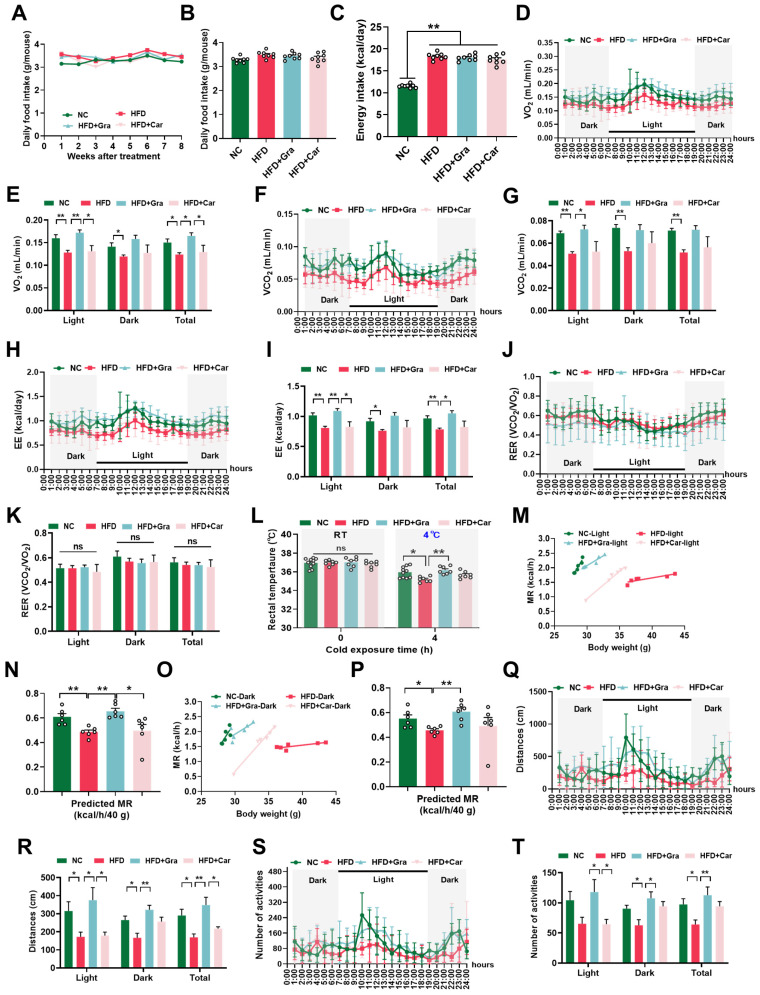
Graphene-FIR therapy has no impact on energy intake but increases EE in HFD mice, (**A**,**B**) Daily food intake of each group. (**C**) Energy intake of mice. (**D**,**E**) O_2_ consumption (V_O2_). (**F**,**G**) CO_2_ release (V_CO2_). (**H**,**I**) Energy expenditure. (**J**,**K**) Respiratory exchange rate (RER). (**L**) Core temperature. (**M**–**P**) Predicted metabolic rate (MR) either in light or dark. (**Q**,**R**) Activity distance within 24 h. (**S**,**T**) Number of activities within 24 h. Data are displayed as mean ± SEM; *n* = 6–10, * *p* < 0.05, ** *p* < 0.01, ns represents no significant difference.

**Figure 6 ijms-26-02225-f006:**
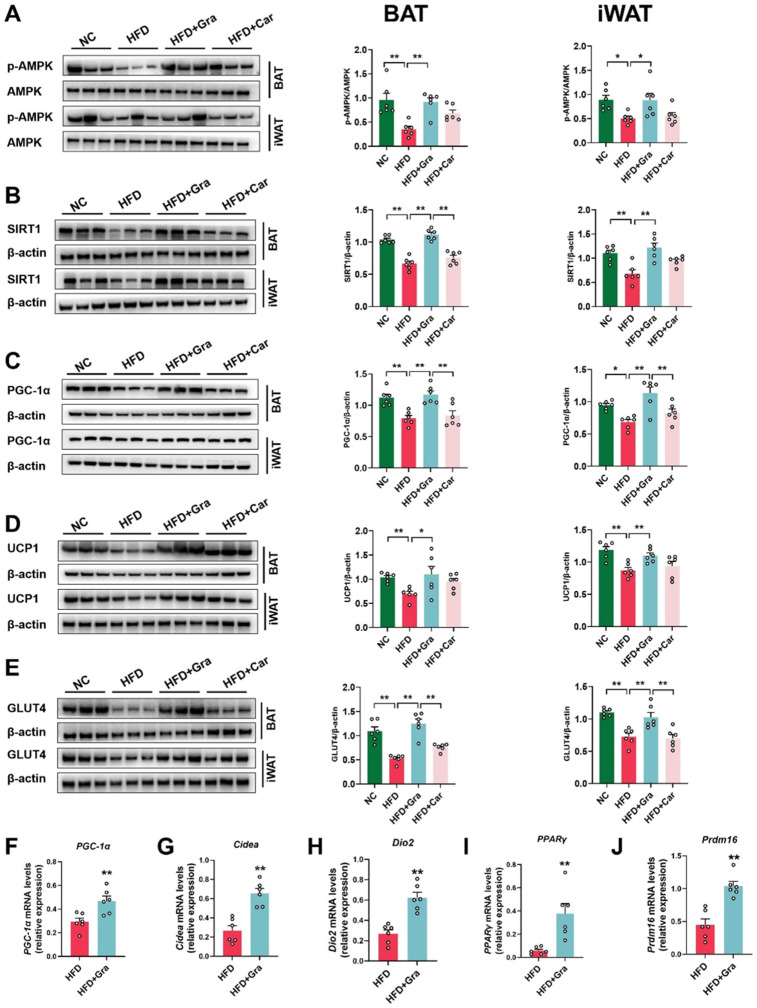
Graphene-FIR therapy could activate adipose tissue thermogenesis and activate the AMPK/SIRT1/PGC-1α pathway in HFD mice. (**A**–**E**) Representative protein bands and relative protein expressions of AMPK, PGC-1α, SIRT1, UCP1, and GLUT4. β-actin was used as a loading control. (**F**–**J**) The mRNA expression of thermogenic genes. Data are displayed as mean ± SEM; *n* = 6, * *p* < 0.05, ** *p* < 0.01.

## Data Availability

The datasets used and/or analyzed in the current study are available from the corresponding author upon reasonable request.

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
