# Peer review of "Graphene-Based Far-Infrared Therapy Promotes Adipose Tissue Thermogenesis and UCP1 Activation to Combat Obesity in Mice"

_ijms, 2025, doi:10.3390/ijms26052225_

Round 1
Reviewer 1 Report
Comments and Suggestions for Authors
The manuscript “Graphene-Based Far-Infrared Therapy promote UCP1 activation and adipose tissue Thermogenesis via AMPK/SIRT1 pathway to combat obesity in mice” demonstrated that graphene-FIR therapy effectively alleviated body fat accumulation, improved dyslipidemia, impaired liver function, and enhanced insulin sensitivity, which are largely dependent on activation of the thermogenic program in adipose tissues. Generally speaking, the study was well designed and written. The data was abundant. The figures were well made.
One major deficiency is the mechanism elucidation. In figure 6, the authors showed the activation of p-AMPK and SIRT1 both in iWAT and BAT under the graphene-FIR therapy. These results only indicated that the activation of p-AMPK and SIRT1 is associated with the graphene-FIR therapy-induction of iWAT and BAT thermogenesis, but could not suggest that “via AMPK/SIRT1 pathway”. All the description about the signaling pathway in the manuscript should be modified, including in the title.
Author Response
Reviewer #1:
- One major deficiency is the mechanism elucidation. In figure 6, the authors showed the activation of p-AMPK and SIRT1 both in iWAT and BAT under the graphene-FIR therapy. These results only indicated that the activation of p-AMPK and SIRT1 is associated with the graphene-FIR therapy-induction of iWAT and BAT thermogenesis, but could not suggest that “via AMPK/SIRT1 pathway”. All the description about the signaling pathway in the manuscript should be modified, including in the title.
Response: Thank you for your valuable feedback. We appreciate your insightful comments regarding the mechanism elucidation in our study. We strongly agree that activation of p-AMPK and SIRT1 is associated with graphene-FIR therapy-induced thermogenesis in iWAT and BAT. We have corrected all the parts involved in AMPK/SIRT1 signaling pathway in the revised manuscript. Thank you again for your constructive comments, which have significantly improved the clarity and rigor of our manuscript.
Reviewer 2 Report
Comments and Suggestions for Authors
The manuscript entitled “Graphene-Based Far-Infrared Therapy promote UCP1 activation and adipose tissue Thermogenesis via AMPK/SIRT1 pathway to combat obesity in mice” provide well-structured experimental work and clearly represented results in support of using graphene-FIR therapy as a comprehensive approach for alleviating HFD-induced metabolic impairment in mice. The results are supported with solid molecular biological techniques that provide mechanistic information for observed effect. The main idea is well-constructed and presented. However, in my opinion, there are several minor issues that should be clarified, prior acceptance for publication.
- Please, perform additional precise proofreading of the text in order to clear some minor stylistic inaccuracies as:
- subscripts (VO2, CO2 -line 24);
- incorrect reference number (line 52);
- some fonts in different color (full stop on line 53);
- Please explain why you’ve mentioned only iWAT in the introduction (line 55-56), or explain in brief in general for WAT and its subtypes if you want to put a highlight on iWAT?
- Provide additional explanation of the type of mimetic agent you mean in line 91-92.
- Find more appropriate phrase, explanation instead of “Successful mice” (line 128).
- Please, doublecheck carefully proofread for English spelling the Supplementary file. Moreover, all tables in Supplementary should be numbered Supplementary Table S1, S2 and so on. Revise the number of the tables in Supplementary materials and methods section, there should be in a total 4 tables and update these numbers in main text, where mentioned.
Author Response
Reviewer #2:
- subscripts (VO2, CO2 -line 24);
Response: We really appreciate your valuable advice. We apologize for not checking the manuscript carefully enough. Following the reviewer's suggestion, we have revised this error in the revised manuscript. The changes are as follows:
Furthermore, graphene-FIR therapy increased VO2, VCO2, and EE levels in HFD mice to exhibit enhanced metabolic activity.
- incorrect reference number (line 52)
- some fonts in different color (full stop on line 53)
Response: We appreciate your suggestion very much. Following the reviewer's suggestion, we have revised this error in the revised manuscript.
- Please explain why you’ve mentioned only iWAT in the introduction (line 55-56), or explain in brief in general for WAT and its subtypes if you want to put a highlight on iWAT?
Response: Thank you for your helpful comments. We have incorporated additional pertinent details into the revised manuscript to provide reviewers and readers with a more comprehensive understanding of our study.
White Adipose Tissue (WAT) is the most important form of fat storage in mammals, and its main function is to store energy and regulate metabolism. WAT is not only a passive energy reservoir, but also an active endocrine organ that secretes a variety of adipokines, such as leptin, adiponectin and resistin. These factors play important roles in energy balance, insulin sensitivity, and inflammatory responses. According to the anatomical location, cellular morphology, and functional differences, WAT can be divided into the following subtypes: inguinal white adipose tissue (iWAT), visceral white adipose tissue (vWAT), and epididymal white adipose tissue (eWAT). Mouse epididymal white adipose tissue (eWAT) is comparable to human visceral white adipose tissue (e.g., omentum, mesentery), whereas mouse inguinal white adipose tissue (iWAT) is comparable to human subcutaneous adipose tissue. iWAT plays an important role in the regulation of metabolism, energy storage, body temperature regulation, and inflammation. Excessive accumulation of eWAT leads to insulin resistance and cardiovascular disease. In this study, we mainly studied the body weight gain and metabolic disorders caused by high-fat diet in mice. Combined with the physiological function of WAT, iWAT was considered to be consistent with the study, so only the function of iWAT was described in the manuscript.
- Provide additional explanation of the type of mimetic agent you mean in line 91-92.
Response: We are very grateful for the reviewer’s comment. We have added relevant information of the revised manuscript so that reviewer and readers can better understand our study.
Metabolic modulators, such as the PPAR agonist GW501516 and the AMP-activated protein kinase (AMPK) agonist, known as "exercise mimetic agents", have been shown to mimic the lipid-lowering effects of aerobic exercise, tricging the body into requiring a higher metabolism to help the body lose weight or improve endurance. Exercise mimetics, compounds designed to replicate exercise benefits, have faced safety issues and ethical controversies. While initially promising for metabolic enhancement, their mis-use in sports has resulted in health complications and subsequent bans by athletic authorities, significantly hindering therapeutic development. In our previous study, graphene-FIR therapy effectively modulated gut microbiota homeostasis and activated AMPK, enhancing physical performance without the side effects associated with drug interventions. It has potential utility as a mimetic agent in treating metabolic disorders.
- Find more appropriate phrase, explanation instead of “Successful mice” (line 128)
Response: Thank you for kindly reminding us. We apologize for the inappropriate description in the previous manuscript. We have revised the description in the revised manuscript.
The mice that met the criteria were divided into a HFD group, a HFD+Gra group, and a HFD+Car group, with 7 mice in each group.
- Please, doublecheck carefully proofread for English spelling the Supplementary file. Moreover, all tables in Supplementary should be numbered Supplementary Table S1, S2 and so on. Revise the number of the tables in Supplementary materials and methods section, there should be in a total 4 tables and update these numbers in main text, where mentioned.
Response: Thank you for the reviewer’s advice. We corrected this part in the revised manuscript. We have added relevant information of the revised manuscript.